

# 1 A modular library for fast prototyping of solution-state nuclear
# 2 magnetic resonance experiments

Michał Górka, Wiktor Koźmiński
Biological and Chemical Research Centre, Faculty of Chemistry, University of Warsaw, Żwirki i Wigury 101, 02-089
Warsaw, Poland
*Correspondence to*: Wiktor Koźmiński (kozmin@chem.uw.edu.pl)
**Abstract.** We present a framework library (Modular Elements, ME) for the development of pulse sequences for Bruker
spectrometers. It implements a two-level abstraction approach–the lower level comprises basic functional elements of pulse
sequences and the higher one often-reused blocks comprising multiple spin echoes. The low-level abstractions reduce code
duplication between variants of experiments such as hard-pulse and selective variants of individual NMR experiments. The
high-level modules enable further reuse of pulse program code and aid in the construction of complex experiments. We show
the library's functionality by presenting pulse programs that can be switched between standard and TROSY variants, hard
and shaped pulses and can seamlessly incorporate real-time homodecoupling. Adaptability is further demonstrated in a
configurable 4D NOESY program.

## 16 1 Introduction

NMR is an extraordinarily powerful and adaptable spectroscopic method, with just the solution-state variant being capable of
discerning the structure and dynamics of molecules ranging in size from simple organic compounds to large protein
complexes such as a proteasome (Sprangers and Kay, 2007). The variety of experimental objects and the great number of
parameters that can be measured has led to the proliferation of not only general experimental schemes (such as an $^{1}$H, $^{15}$N
HSQC (Bodenhausen and Ruben, 1980) or a HNCO (Kay et al., 1990b; Ikura et al., 1990)), but also their variants and thus
the pulse sequences implementing them as computer code. As an example, for the oft-used HNCO experiment, the non-
exhaustive list of meaningful implementation choices is: the experiment can use hard pulses or avoid saturating water using
selective pulses (Schanda et al., 2006); the final transfer element can be a simple spin-echo (Palmer et al., 1991), a set of
three echoes implementing a sensitivity-enhanced transfer or one of many TROSY variants (Salzmann et al., 1999b;
Nietlispach, 2005), with possible optimizations (Salzmann et al., 1999a; Schulte-Herbrüggen and Sørensen, 2000); radiation
damping can be suppressed with bipolar gradients (Sklenar, 1995). Even without implementing all specialized experiment
variants, the standard library supplied with the TopSpin software (Bruker) contains over a thousand pulse programs.



A common problem with pulse sequences, especially in biological NMR, is thus the requirement to code multiple variants of
a given sequence. If this is done in separate files (as in the TopSpin built-in library) it results in a lot of code repetition and if
made using conditional statements, it can substantially complicate the structure of the file, making trouble-shooting harder.
Similarly, many pulse sequences share large amounts of code often with no or minimal changes. Because this repeated code
is scattered across different sequences and variants of experiments adding new variants (using different soft pulses, adding
homodecoupling) requires applying the same modification across a large part of the whole pulse sequence library, which is
tedious and error-prone. It is possible to implement such a library using standard systems programming language like C or
Python, but we decided to use the native programming language of the spectrometer system, since any user writing pulse
sequence needs to be familiar with it and requiring knowledge of separate programming language and its tooling would be
an unnecessary hurdle to adoption. Here we show that by abstracting certain functionality using the somewhat limited macro
and "define" functionality built into the TopSpin software the above-described problems can still be avoided and the code
can be made more readable and easier to modify. Here we present the Modular Elements (ME) library for Bruker
spectrometers. Although the library is specific to a particular hardware vendor, the modular approach it implements is more
general and can be implemented on other instruments. A previous implementation of a modular library for pulse program
implementation (NMR blocks) can be found in (Zawadzka-Kazimierczuk, 2012) for Varian/Agilent spectrometers, where
spin echos and transfer periods such as INEPT or COS-INEPT where abstracted as C functions. Alternative approaches to a
modular library include domain specific pulse program generators, like GENESIS (Yong et al., 2022) for NOAH
supersequences. Specialized libraries combining custom pulse programs and various tools (Favier and Brutscher, 2019;
Vallet et al., 2020; Lukavsky and Puglisi, 2001), are suitable for routine use, but have limited applicability in the prototyping
of new sequences.
**2 General approach to pulse sequence modularisation**
We categorise the library's functionality as low- and high-level. Low-level functionality encompasses the creation of
variables and functions (technically functional macros), abstracting the basic building blocks of pulse sequences like pulses,
gradients and delays. A pulse function can evaluate to a 90 degree proton hard pulse or an HN selective excitation pulse
depending on global settings. A gradient function can evaluate to "no operation" in standard HSQC, or a selection gradient it
the gsHSQC variant, with its corresponding delay variable containing a zero or correspondingly non-zero length of time.
Decoupling functions for protons and deuterium can likewise be enabled and disabled depending on whether a TROSY
variant is desired and if the sample is deuterated. This functionality simplifies the writing of pulse sequences implementing
multiple variants of a given NMR experiment and gives its user the ability to easily test and compare the effectiveness of the
variants for a given sample and the commonization of parameters across variants enables faster optimisation.
High-level functionality is implemented as modules that are included whole in the pulse sequences and can be classified as
general modules and specific modules. General modules implement elements common to almost all pulse sequences. The



functionally most significant ones are the preparation and acquisition modules. The preparation module gives the user the
option to turn on functionality such as solvent presaturation or a combination of N/C pulses and pulse field gradients for
spoiling of residual magnetisation on those nuclei. The acquisition module enables switching between standard or
homodecoupled data acquisition. The specific modules are abstract blocks of pulse sequence elements that appear in many
pulse sequences in an almost identical form. Two main types of specific modules are proximal and distal modules,
abstracting the functionality of blocks including and following first excitation (distal) and right before acquisition
(proximal). Despite a large variety of possible implementations, the proximal/distal fragments differentiate variants of a
pulse sequence (for example a standard hard-pulse HNCO, selective/BEST-HNCO, hard pulse and BEST TROSY-HNCO
(Solyom et al., 2013)) and the actual code is usually repeatable across different sequences. HNCO, HNCACO and HNCOCA
(Yang and Kay, 1999) have very similar proximal and distal parts; HN(CA)CONH and HabCabCONH (Kazimierczuk et al.,
2010) have different proximal blocks, but the distal block is still very similar for all sequences listed. With the use of the
low-level functionality described above a single proximal module can abstract the initial two transfer periods (first with
transverse H magnetisation and second with transverse N/C magnetisation), with the choice of N/C nucleus and choice of
evolved J coupling (CO in HNCO) made using define directives in the main pulse sequence. NOESY experiments are
particularly susceptible to modularisation, with the NOE transfer period naturally splitting them into proximal and distal
blocks. Standard 2D experiments of the HSQC, TROSY and HMQC type have thus been implemented as proximal modules,
that can be used on their own as 2D experiments or included in a 3 or 4D NOESY (Kay et al., 1990a) with the chosen distal
modules, which can themselves be modified 2D experiments or simpler blocks.

## 3 Library implementation

Description of implementation details and design choices requires a quick recapitulation of TopSpin pulse programs
language specifics. TopSpin allows has two types of variables: user-adjustable numbered variables (*d1..d63* for delays,
*cnst1...cnst63* for floating point constants, similarly for integer constants ("loopcounters") *lN*, pulse lengths *pN*, ...) and
named variables (pulses, delays and loopcounters only, also lists of various kinds), which can only be manipulated within a
pulse program. Some less-documented observations on the limitations of named variables are compiled in SI. TopSpin
implements limited functionality for defining text-substitution macros ("--traditional" mode of the GNU C preprocessor cpp
(Stallman and GCC Developer Community, 2012)), which can be used everywhere outside a "relation" (variable value
calculations using a subset of C syntax), due to their implementation as text in quotes (treated as string literals by cpp and
ignored for macro expansion), though this limitation can be overcome (see the file "notes on TopSpin.txt" in the ME library).
The user can provide custom option choices to a pulse program using the ZGOPNTS variable to define appropriate macros.



### 3.1.1 Low-level modularisation

### 3.1.1 Variables

With no user-adjustable named variables, two approaches to making them consistent across different pulse programs are possible - indirection through a named variables or introducing a convention attaching constant meaning to numbered variables. Due to the limited number and type of named variables we predominantly use the latter option (with sets of variables described in files such as delays.incl, pulse.incl, ...) with some focused use of indirection - for example proximal type modules use *timeHX* and *timeXY* for J coupling evolution times between the H, X and Y nuclei. For variables that don't ordinarily have calculations performed on them (pulse phases *phN*, gradient programs *gpN*) with implemented full indirection, where the user can use phFree1 or phFree3 without worrying as to which phN variables are used by other parts of a pulse program.

### 3.1.2 Pulses

The most important low-level abstractions are pulse functions. They are implemented using function-like macros of cpp and have the general form of nucleus_type(phase), where nucleus can be a general specifier like H/C/N or more specific like HN/HC/CA/CO and type is classified based on the desired functionality, with the main ones being: excitation (for the excitation of longitudinal magnetization), flipback (acting on transverse magnetization), refocussing, inversion (inverting longitudinal magnetization), excitation_UR and flipback_UR (implementing universal rotations). The pulse macros will have different replacement text based on global settings (usually ZGOPTNS). A proton pulse "H_excitation(ph)" will evaluate to a hard pulse "p1 ph pl1" by default, but with a "-DH_SHAPED" option will instead evaluate to "p54:sp54 ph" for a selective soft pulse and the associated named variable pH_excitation will be set to have the same value as *p1* or *p54*.

Pulse programs should account for the effective evolution time a during pulse (which can be as much as 1 ms for longer selective pulses) to give correctly phased spectra and optimal J coupling evolution times. This library only accounts for linear phase slope using the modelling method described in (Lescop et al., 2010), that is treating a pulse as sequence (delay, ideal pulse, delay), which accounts for the phase slope of many commonly used pulses and can be optimized for consciously during pulse design (Gershenzon et al., 2008; Asami et al., 2018). This phase slope is compensated for using variables such as *eH_excitation*, which for the hard pulse above would be set to $\frac{2p1}{\pi}$. We assume that the flipback and flipback_UR pulses act as if they were time-reversed excitation pulses and so the effective evolution time for a flipback pulse acting on transverse magnetization is also *eH_excitation*. For a H_excitation_UR of phase x will give an effective time of *eH_excitation* for z magnetization, *eH_flipback* for y magnetization and *eH_excitation + eH_flipback* for x magnetization. By compensating delays using the above mentioned variables the whole sequence can be switched from a hard pulse implementation to a shaped pulse version, whether to account for field inhomogeneity or perform band-selective excitation.





### 3.1.3 Code blocks

There are many small blocks of code that can be included/excluded in a pulse program based on a sequence variant. To limit
the number of conditional statements in the main pulse program, many are defined as macros that evaluate to pulse program
code based on options, for example "H2O_FLIPBACK(ph2)" ill evaluate to "(11:sp1 ph2):f1" or a pulse sequence with
water flipback and to whitespace if using selective pulses. Similarly DECOUPLE_H_ON and DECOUPLE_H_OFF macros
will turn on proton decoupling in a standard HNCO experiment but will have no effect in TROSY-HNCO.

### 3.2 High-level modularization

TopSpin pulse programs follow a defined sequential structure that complicates the implementation of high-level modules as
individual files and, in general, is:

1) configuration and compile-time calculations

2) a "zd" or "ze" statement

3) pulse program body (pulses and delays) and real-time calculations

4) signal acquisition block

5) loop statements for scans of a FID and points of a multidimensional experiment

6) phase program definitions

### 3.2.1 General modules

The general modules fit into this sequential structure as follows:

1a) configuration and compile-time calculations

1b) **init. incl**

1c) configuration and compile-time calculations continued

2) a "zd" or "ze" statement

3a) real-time calculations

3b) **start.incl**

3c) pulse program body (pulses and delays) and real-time calculations

3d) **end.incl**

5) loop statements for scans of a FID and points of a multidimensional experiment

6a) **phasecycles.incl**

6b) phase program definitions

The general modules have numerous conditional statements and imports evaluating the option provided in point 1) above
and using the built-in ZGOPTNS variable and interact with the specific modules (this is covered below). The init.incl
module provides the libraries core functionality by defining macros for functions and variable descriptions. start.incl



executes the relaxation delay (with possible solvent presaturation) and optional operations, such as crushing residual C or N
magnetization (gradient pulse after an excitation pulse) or inverting N magnetization before the relaxation delay in BEST-
TROSY. For non-protein experiments an ASAP (Kupče and Freeman, 2007) period would be added here, but the relevant
code is experimental and provided in a commented-out form due to the method's potential to damage probeheads. The
end.incl module handles acquisition with the option for real-time homodecoupling - here provided with $^{13}$C-GBIRD$^{r,X}$
(Garbow et al., 1982; Haller et al., 2022) and BASHD (Brüschweiler et al., 1988; Krishnamurthy, 1997) types.

### 3.2.2 Specific modules

In contrast to the general modules, specific modules implement a specific form of proximal or distal block and serve to
localize the relevant code in a single file. The biggest hurdle to writing self-contained modules for TopSpin is the sequential
pulse program structure necessitating the separation of related code segments in the post-preprocessing file. To mitigate this
problem each module is entirely enclosed in a conditional statement with alternative conditions (an if...elif...else structure)
and including the file once will only insert a selected part of the module into a file. Since the 4 general modules already
perform the sequential separation of code each of them sets the appropriate conditions (defines a macro) and imports the
distal_2D.incl and proximal_2D.incl which themselves import the selected specific modules at each of the 4 positions in the
pulse program. Thus, the initialization phase statements (variable declarations, some calculations, macro definitions) are
included in init.incl, runtime calculations of both types of modules are included through start.incl, together with the main
body (pulses and delay statements) of the distal. Similarly, the main body of the proximal module is included through the
end.incl before the latter's acquisition portion. Phase cycles of both modules are inserted into a pulse program file through
phasecycles.incl with some basic logic allowing for coordinating the cycles between them if two modules are used.
For triple-resonance experiments (in the implementation limited to amide protons, but should be possible to extend to
aliphatic/aromatic groups) the proximal module hx.incl and the distal module hx.incl provide the ability to compartmentalize
the relatively standard blocks for both out-and-back and straight through type experiments and a more detail description in
the context of a HNCO experiment is provided below. Although sub-optimal in some circumstances the library provides a
default 2 step phase cycles for each of the modules, leaving the implementation of 8 step and longer cycles for the central
part of the program.
A specific module separate from the proximal-distal type can also be based on the same structure and either manually
included in the pulse program after each general module or in a specific module itself - se.incl is module implementing the
sensitivity-enhanced COS-INEPT and TROSY transfers and is imported in both the hsqc_se.incl and hx.incl modules.



**4 Application examples**
**4.1 HNCO**

```
prosol relations=<me>

# include <Avance.incl>
# include <Grad.incl>

"in1 = inf1"
"in2 = inf2"
define delay T1
"T1 = 0"
define delay TPmax
"TPmax = max(in2*(td2/2 - 1), 0)"

# define XH
# define HX
# define DISTAL_N
# define DISTAL_Y_CO
# define DISTAL_A_CA
# define PROXIMAL_NH
# define PROXIMAL_Y_CO
# define PROXIMAL_A_CA

# include <ME/includes/init.incl>

1 ze

# include <ME/includes/start.incl>

; COzNz CO evolution (T1):
  (CO_excitation(phFree1)):fCO
  T1*0.5
  (center (CA_CO_inversion(ph0)):fCA (N_inversion(ph0)):fN)
  T1*0.5
  (CO_refocussing(ph0)):fCO
  (CA_CO_inversion(ph0)):fCA  ; BSP compensation.
  (CO_flipback(ph0)):fCO

GRAD(gpFree1)

# include <ME/includes/end.incl>
  d11 mc #0 to 2
    F1PH(calph(phFree1,+90), caldel(T1, +in1))
    PROXIMAL_MC2
exit

# include <ME/includes/phasecycles.incl>

phFree1 = 0 0 0 0 2 2 2 2

; Receiver phase:
ph31 = PROXIMAL_PH31 + DISTAL_PH31 + phFree1

;gpzFree1: gradient after CO echo: 21%.

;gpnamFree1: SMSQ10.100
```

**Fig. 1. Pulse program code for the implementation of the HNCO experiment.**





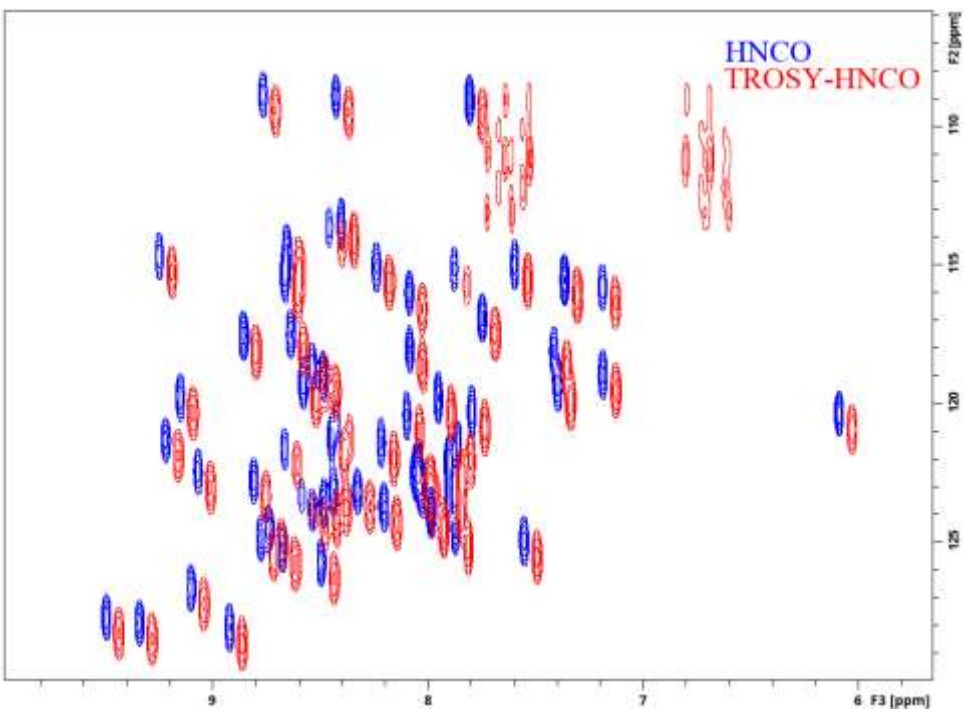

**Fig. 2. Experimental demonstration of the implementations of the HNCO and TROSY-HNCO experiments for ubiquitin 8 (kDa) at 25 °C. Spectra were recorded as $^1$H-$^{15}$N planes with maximum evolution times of 85.2 ms ($^1$H) and 9.87 ms ($^{15}$N) and processed using cosine squared window functions.**

HNCO is one of the simplest triple-resonance experiments and thus a good candidate to demonstrate the strengths and limitations of the presented approach to library building. We present its ME NMR implementation in Fig. 1. We use a custom prosol file (used mostly for automatic precalculation of pulse parameters) to free up a number of variables. Evolution delays and increments are defined explicitly due to the proximal module's numbered variables (here td2 and in2) being dimensionality-dependant. The block of defines specifies options for ME library - specifying the proximal (xh.incl) and distal (hx.incl) modules and the couplings to be evolved (Y is $^2J_{NCO}$) and decoupled (A is $^2J_{NCA}$). After importing the first two general modules, which includes the distal modules two spin echoes, the carbonyl echo is implemented using the library's low-level functionality. Since the channels and pulses aren't selected explicitly the sequence this block will function with split CA and CO channels (with the right spectrometer configurations and "CACO_SPLIT" defined in ZGOPTNS). The rest of the pulse program includes the end.incl module (with the two proximal echoes and acquisition) and standard configuration of gradients and phasecycles. To demonstrate the libraries functionality in Fig 2. we present 2D spectra (recorded as HN(CO) experiments) of a standard variant of the experiment (no ZGOPTNs) and a TROSY-HNCO (adding the TROSY define to ZGOPTNS) selecting only the $H_\beta$ and $N_\beta$ component (the lower right component using standard display convention). It's possible to choose a $^{13}$C-GBIRD$^{r,X}$ appending the "ACQ_BIRD_C" option to ZGOPTNS, with an example of line narrowing demonstrated in Fig. 3.



**Fig. 3.** 1D slices (for $N = 128.5$ ppm) through $^{1}$H-$^{15}$N planes recorded for a TROSY-HNCO with standard acquisition and
TROSY-HNCO with $^{13}$C-GBIRD$^{r,X}$ demonstrating the effectiveness of the homodecoupling and the resultant line narrowing. Both
spectra were acquired for ubiquitin 8 (kDa) at 25 °C with maximum evolution times of 340.7 ms ($^{1}$H) and 9.87 ms ($^{15}$N) and
processed using a cosine squared window function in the N dimension and sine squared shifted by $\frac{\pi}{\Box}$ in the H dimension. The
GBIRD spectrum was shifted right by 4 Hz (shift was possibly induced by sample heating) and scaled up to match the amplitude of
the standard TROSY-HNCO. For the GBIRD spectrum 18 chunks were acquired with a 11.96 ms inter-chunk delay, 3.5 ms $^{2}$J$_{HC}$
evolution time and using a 120 μs BIP-720-100-10 (Smith et al., 2001) pulse for $^{13}$C inversion. Linewidths at half height are (from
left to right) 19.6 Hz, 19.5 Hz and19.1 Hz for the standard spectrum and 13.2 Hz, 13.2 Hz and 13.7 Hz for the homodecouple
spectrum (TopSpin peakw function).



**4.2 4D NOESY**

```
prosol relations=<wt>

# include <Avance.incl>
# include <Grad.incl>

# include <WT/includes/init.incl>

"in1 = inf1"
"in2 = inf2"
"in3 = inf3"

define delay mixTime
;d10: NOESY mixing time [40-400 ms]
"mixTime = d10 - pGRAD - dGRAD" ; Corrected for gradient.

1 ze

# include <WT/includes/start.incl>

; NOESY mixing:
  mixTime
  GRAD(gpNOESY)

# include <WT/includes/end.incl>

  d11 mc #0 to 2
    DISTAL_MC1
    DISTAL_MC2
    PROXIMAL_MC3
exit

# include <WT/includes/phasecycles.incl>

; Receiver phase:
ph31 = PROXIMAL_PH31 + DISTAL_PH31

;gpzNOESY: gradient after NOESY: -7%.
;gpnamNOESY: SMSQ10.100
```


**Fig. 4. Pulse program code for the implementation of a 4D NOESY experiment.**
The modular nature of the library is exemplified by the 4D NOESY pulse program in Fig. 4. Apart of the basic structure
described above in the case of HNCO it only contains a mixing period joining the proximal and distal module, with the
evolved heteronuclei and experiment types selected by the user using ZGOPTNS. A HC,NH-HMQC-NOESY-HSQC with
sensitivity enhancement in the last dimension (Fig. 5.) can be changed to a HC,CH-HMQC-NOESY-HSQC (Fig. 6.) pulse
program by changing the "PROXIMAL_N" option to "PROXIMAL_C"and adding the gradient selection option ("S", which
isn't a default for non-sensitivity-enhanced HSQC.



**Fig. 5. $^1$H-$^1$H planes recorded using a 4D HC,NH-HMQC-NOESY-HSQC experiment for ubiquitin 8 (kDa) at 25 °C. Spectra were**
**recorded with maximum evolution times of 85,2 ms ($^1$H direct dimension) and 6.99 ms ($^1$H indirect dimension) and processed using**
**cosine squared window functions.**

**MAGNETIC RESONANCE**
Discussions

**228**
**229** **Fig. 6. Two different $^{1}$H-$^{13}$C 2D planes recorded using a ME implementation of a 4D HC,CH-HMQC-NOESY-HSQC experiment**
**230** **for ubiquitin 8 (kDa) at 25 °C. Spectra were recorded with maximum evolution times of 85.2 ms ($^{1}$H direct dimension) and 7.96 ms**
**231** **(both $^{13}$C dimensions) and processed using cosine squared window functions.**



### 4.3 $^1$H-$^{15}$N correlation – shaped pulses

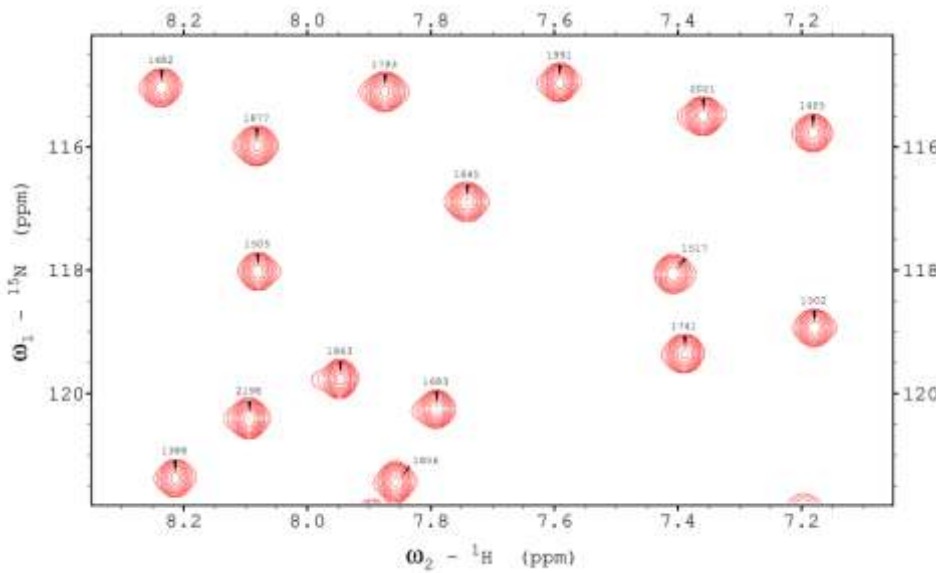


**Fig. 7. $^1$H,$^{15}$N TROSY spectrum recorded using a ME implementation with hard pulses and water flipback for ubiquitin 8 (kDa) at 25 °C. The spectrum was recorded with maximum evolution times of 85,2 ms ($^1$H) and 39.5 ms ($^{15}$N) and processed using cosine squared window functions.**

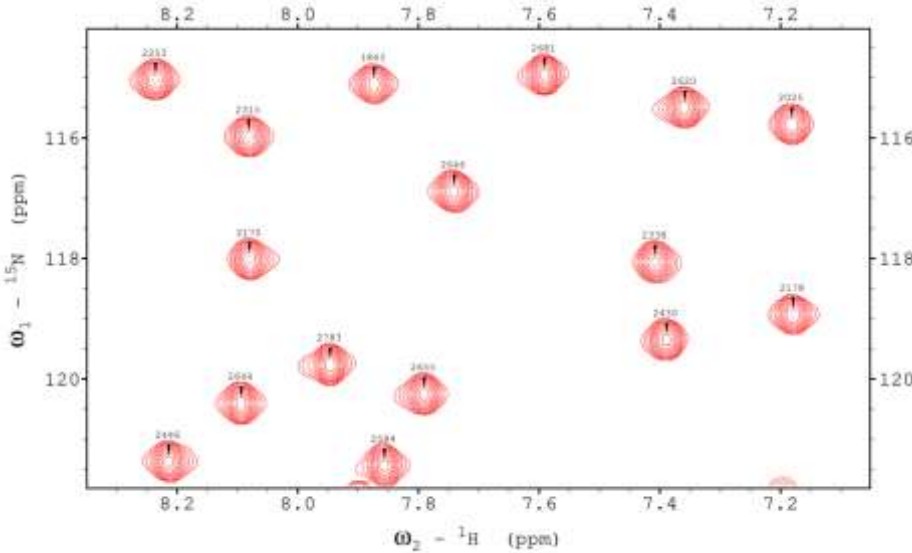


**Fig. 8. Fig. 7. $^1$H,$^{15}$N TROSY spectrum recorded using a ME implementation with shaped pulses (E400B and RE-BURP) for ubiquitin 8 (kDa) at 25 °C. The spectrum was recorded with maximum evolution times of 85,2 ms ($^1$H) and 39.5 ms ($^{15}$N) and processed using cosine squared window functions.**




Since BEST-type experiments utilizing shaped pulses can bring improved sensitivity even at higher scan repetition rates []
we demonstrate the library's inherent ability to automatically adapt to the substantial chemical shift and coupling evolution
during the 90-degree universal rotation E400B (Veshtort and Griffin, 2004) (using a time-reversed version of the original
pulse for excitation) pulses with the length of 1073.1 us (equivalent to an ideal pulse followed by a 611.7 us delay) and
refocussing pulse RE-BURP (Geen and Freeman, 1991)  with length of 1108.8 us (modelled as an ideal refocussing pulse
flanked by 554 us delays) in Fig. 7. and 8. With a relaxation delay of 0.65 s all peaks in the selected region are over 20%
stronger in the shaped pulse version.  Full datasets for a number of different relaxation delays are provided as in the data
availability section.

## 5 Materials & methods

For all experiments we used a 2 mM $^{13}$C, $^{15}$N-double labelled human ubiquitin (ASLA Biotech) in a 5 mm Shigemi NMR
microtube. All spectra were acquired using a Bruker Avance IIIHD 800 MHz spectrometer with a 5 mm TCI z-gradient
cryo-probe. Pulse lengths for 90 degree hard pulses were 10.47 μs for $^{1}$H, 12.3 μs for $^{13}$C and 33.22 μs for $^{15}$N. Full
acquisition and processing parameters are provided in the dataset linked below in the Data availability section. Acquisition
and library testing was performed using the TopSpin 3.6.5 Service Pack 2 software (Bruker). Data processing and plotting
(aside from Fig. 7. and 8.) was carried out in TopSpin. Figures 7 and 8 were prepared using the NMRFAM-SPARKY
software (Goddard and Kneller, 2004; Lee et al., 2015).

## 6 Conclusions

We have described a framework library implementing a two-level approach to pulse program modularization and
demonstrated its utility. We hope it can be used by others either directly for the streamlining of pulse program code or as an
inspiration for similar frameworks. Although the usefulness of the modularization approach is most obvious for the case of
protein experiments presented here it should extend to nucleic acids and, to a more limited extent, small molecules. In the
latter case the ability to modularize preparation period operations (presaturation, ASAP), WATERGATE (Piotto et al., 1992;
Sklenar et al., 1993) type solvent suppression and real-time acquisition should be particularly useful.

## Code availability

The initial version of the ME library is available online at: https://doi.org/10.5281/zenodo.10578681. Current library version
is available from the authors upon request.



**Data availability**

All data used in the preparation of this article is available online at: https://doi.org/10.5281/zenodo.10578330.

**Author contributions**

MG and WK designed the general workflow of the ME library. MG wrote the library code and performed the experiments. MG wrote the manuscript with input from WK.

**Financial support**

This research was supported by the Polish National Science Centre grant PRELUDIUM 2015/19/N/ST4/00863 to MG.

**Competing interests**

The authors declare that they have no conflict of interest.

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
