# Peer review of "A modular library for fast prototyping of solution-state nuclear 2 magnetic resonance experiments"

_Magnetic Resonance, 2024_

## Author Response (AR1)

**RC1**

**Comment:** My personal critique is that the proposed library statements in some cases are not easily readable or memorable, which may repel some potential users. For instance, the code for the HNCO experiment (Fig. 2) does make some sense whereas the 4D NOESY pulse program syntax (Fig. 4) hardly makes any sense. It starts with a mixTime statement which may include some pulses, but as the first statement in a NOESY experiment makes little sense.

**Response:** We agree that parts of the library syntax aren't particularly readable and might obscure its functionality. Since the most straightforward method of using the library to implement a new pulse program is to copy and edit an existing one, we will improve their readability.

**Improvements to the manuscript:** We added comments to the sequences describing the functionality of imports, most significantly the use of start.incl and end.incl modules to select 2D blocks in NOESY programs and updated Fig. 1. and Fig. 4.

**Comment:** A couple lines down the sequence this is followed by DISTAL_MC1, DISTAL_MC2 and PROXIMAL_MC3. Again, these statements make little sense as such, but it is also difficult to see whether more user-friendly syntax can be invented.

**Response:** The MC macros appeared in the pulse program code due to the limitations of the programming language. The approach used in the original submission was a compromise between user-friendliness and customizability. Upon deeper reflection we have decided to separate the programming facilities for both scenarios. We will revise the text to use a user-friendly version and describe the customizable (expert) facilities in library documentation.

**Improvements to the manuscript:** We changed the library code, so that those statements aren't needed in standard pulse programs, and updated the corresponding figures (Fig. 1. and Fig. 4).

**Comment:** Finally, it is also not clear whether the Bruker TopSpin software can display such sequences for users to understand the experiments in detail.

**Response:** TopSpin is able to render a pictorial representation (using GUI or the spdisp command) of the pulse program based on the current value of the ZGOPTNS parameter. Unfortunately, the same cannot be said for the configuration of variants using ZGOPTNS as there is no default GUI facility for presenting possible options or indicating which are mutually exclusive. We implemented an experimental GUI configurator utility for the setting of common ZGOPTNS options using radio buttons and drop-down menus. Since the tool relies on a poorly documented API and is prone to cause unintended consequences we have decided against mentioning it directly in the manuscript.

**Improvements to the manuscript:** We have added an experimental configuration utility to the library code with a short description in the supplement.

**Comment:** Minor remarks:

1) Introduction, line 6 - "oft-used" should be spelled out properly.

2) Page 2, section 2 - "... function can evaluate to a 90 degree ..." perhaps would better read "function can
translate into a 90 degree ...". See other similar instances in the main text.

3) Page 14, line 1 - a reference is missing.

**Response:** A number of linguistic errors have unfortunately slipped through our initial proof-reading. We will
carefully review the language used throughout the manuscript.

**Improvements to the manuscript:** We have corrected a number of language errors:

oft-used to often used for cpp macros we have replaced "evaluate to" with "expand to" or "replaced by", as per cpp documentation.

and added the missing reference to Schanda, P., Van Melckebeke, H., and Brutscher, B.:

Speeding Up Three-Dimensional Protein NMR Experiments to a Few Minutes, J. Am. Chem. Soc., 128, 9042–
9043, https://doi.org/10.1021/ja062025p, 2006.

RC2

**Comment:** Running a NMR experiment requires an appropriate pulse sequence as well as a related set of
acquisition parameters (pulse lengths, pulse shapes, frequency offsets, delays, constants, …) . As I understand
some of these parameters are predefined, but others need to be set by the user. How is this handled in practice?

**Response:** The experiments use a combination of automatically set (using prosol utilities) or calculated
parameters. All user-set parameters are accessible from the appropriate TopSpin menu (accessed using the ased
command) and are annotated with a description and a default value. In the library we provide a TopSpin utility
for setting default values of parameters defined within the library, which can be used to configure individual
parameter sets or used to prepare a base parameter set that can then be copied and re-used for multiple
experiments. Unfortunately, TopSpin parameter sets aren't usually re-usable between different spectrometers.
Although the parameter sets can be relatively easily adjusted, they also pose a risk to the spectrometer if some of
the adjustments are omitted and inappropriate parameter values or routing configuration is used. For this reason,
we chose not to directly distribute parameter sets.

**Improvements to the manuscript:** We have added a mention of the parameter-setting program in the section
3.1.1 (lines 97-98).

**Comment:** Along the same lines, making these modules usable by a broad range of users requires a
documentation of all available modules, what they are doing, what are the available options and how to
implement them, as well as the relevant acquisition parameters that need to be adjusted. I could not find such a
compiled description ?!

**Response:** We agree that the library requires separate documentation, especially since TopSpin's pulse
programming language and associated environment don't have any built-in helper facilities like docstrings and
syntax completion. Initially, we had problems with finding an appropriate format - especially one that would
allow for easy modification as the library evolves. We have now made the choice to provide the documentation
using Markdown, making it directly accessible from the libraries GitHub repository without the need to use
external software and easy to update using a just a text editor.

**Improvements to the manuscript:** We have added documentation to the library repository provided in the
"Code availability section", with a short mention in section 3.2.2.

**Comment:** As also pointed out by Eriks Kupe, the examples given (HNCO and 4D NOESY) in the manuscript
are certainly very compact in terms of lines of code, but difficult to understand as such. Again it would be useful
to somehow add comments that make the pulse code more readable. I doubt that many potential users will dig
deep into the modular elements code to understand what they are doing.

**Response:** We concur that parts of the initial codebase were too terse. Since the primary interaction users have
with the library is through a pulse program, we will improve the readability of the included example
experiments.

**Improvements to the manuscript:** In the library code, we will include comments that expand on the
functionality of the imports used in the supplied pulse programs. We have updated the figures 3. and. 4.
illustrating HNCO and NOESY sequences.

**Comment:** May be a useful addition to the manuscript would be to pick out 1 or 2 elements, and discuss in a
separate section what spin manipulation they are performing, what options they provide and how they can be
implemented, and what other parameters need to be taken care of when using this particular module.

**Response:** In the initial submission, we neglected to show in detail, how the concepts of general and specific
modules are implemented. Due to the limited programming facilities available even the simplest module
(hsqc.pp) is relatively long and verbose. We think that putting a detailed description directly in the manuscript
would hamper its readability and provide the description in the supplement.

**Improvements to the manuscript:** In the supplement we have added a section describing the functionality of
the 2D.me.pp pulse programe and the hsqc.pp specific module in detail. It explains how a 2D HSQC experiment
can be implemented using ME library's low-level functionality and covers options for gradient selection, water
flipback, bipolar gradients and accommodation of shaped pulses.

**Comment:** The authors employ at several occasions the term "multiple spin echos" to refer to particular
coherence transfer pulse sequence blocks. I suggest replacing this by more commonly used nomenclature ?!

**Response:** We used the term spin echo to describe a part (block) of a pulse program with 180 deg pulse flanked
by delays and 90-degree pulses. We will change this to "evolution periods" and we will reconsider each use of
this term in the text separately, to preserve clarity.

**Improvements to the manuscript:** We changed all the uses of the offending phrase to "evolution periods".

**Comment:** Page 14, line 242: I would replace "even at higher scan repetition rates" by "especially at higher scan repetition rates". And the reference is missing.

**Response:** We have reworded this sentence fragment as suggested, as it originally carried the wrong emphasis. In our experience, BEST experiments were almost always more sensitive and the effect was more pronounced at higher scan repetition rates.

**Improvements to the manuscript:** We changed the wording as suggested and added the missing reference to Schanda, P., Van Melckebeke, H., and Brutscher, B.: Speeding Up Three-Dimensional Protein NMR Experiments to a Few Minutes, J. Am. Chem.Soc., 128, 9042–9043, https://doi.org/10.1021/ja062025p, 2006.

**Comment:**

- Please avoid Bruker nomenclature in the main text, e.g. gsHSQC (page 2, line 54)

-There are quite some typos in the manuscript. Some examples are listed below:

Page 2, line 53: "it" instead of "in"

Page 2, line 59: "whole in" ?

Page 3, line 81: "allows has"

Page 4, line 93: "a named variables"

Page 4, line 109: "time a during pulse"

Page 4, line 112: " for consciously during"

Page 8, line 194: "the sequence this block"

**Response:** We have removed the jargon terms and corrected a number of language errors.

**Improvements to the manuscript:** We have corrected all of the listed errors and a number of others.

RC3

**Comment:** Indeed, I would agree with Bernhard Brutscher that a more detailed description of the proposed library functions including the graphical representations of pulse schemes of the modules and modular schemes of the experiments / pulse sequences would be useful and I would expect this to become available in the associated library manual, which hopefully exists in one form or another. Likewise, for users more interested in the pulse sequences rather than the library functions a sub-section with the most frequently used experiments would be equally useful. These are, of course suggestions for future improvements of otherwise a great idea.
**Response:** We have added library documentation using Markdown, making it directly accessible from the libraries GitHub repository.

**Improvements to the manuscript:** We have added documentation to the library repository provided in the "Code availability section", with a short mention in section 3.2.2.

**Minor corrections not in comments:**

Section 3.1 was incorrectly labelled as 3.1.1 in the initial submission.

In Section 3.2.1 point 4 was incorrectly labelled as 3d).

In the description of HNCO in section 4.1 the sentence on the carbonyl echo was missing its second part, describing its function in standard spectrometer configuration (single carbon channel).